# In Doxorubicin-Adapted Hodgkin Lymphoma Cells, Acquiring Multidrug Resistance and Improved Immunosuppressive Abilities, Doxorubicin Activity Was Enhanced by Chloroquine and GW4869

**DOI:** 10.3390/cells12232732

**Published:** 2023-11-29

**Authors:** Naike Casagrande, Cinzia Borghese, Michele Avanzo, Donatella Aldinucci

**Affiliations:** 1Division of Molecular Oncology, Centro di Riferimento Oncologico di Aviano (CRO), IRCCS, 33081 Aviano, Italy; cp.borghese@libero.it (C.B.); daldinucci@cro.it (D.A.); 2Department of Medical Physics, Centro di Riferimento Oncologico di Aviano (CRO), IRCCS, 33081 Aviano, Italy; mavanzo@cro.it

**Keywords:** Hodgkin lymphoma, doxorubicin, drug resistance, cross-resistance, immunosuppression

## Abstract

Classical Hodgkin lymphoma (cHL) is a highly curable disease (70–80%), even though long-term toxicities, drug resistance, and predicting clinical responses to therapy are major challenges in cHL treatment. To solve these problems, we characterized two cHL cell lines with acquired resistance to doxorubicin, KM-H2dx and HDLM-2dx (HRSdx), generated from KM-H2 and HDLM-2 cells, respectively. HRSdx cells developed cross-resistance to vinblastine, bendamustin, cisplatin, dacarbazine, gemcitabine, brentuximab vedotin (BV), and γ-radiation. Both HDLM-2 and HDLM-2dx cells had intrinsic resistance to BV but not to the drug MMAE. HDLM-2dx acquired cross-resistance to caelyx. HRSdx cells had in common decreased CD71, CD80, CD54, cyt-ROS, HLA-DR, DDR1, and CD44; increased Bcl-2, CD58, COX2, CD26, CCR5, and invasive capability; increased CCL5, TARC, PGE2, and TGF-β; and the capability of hijacking monocytes. In HRSdx cells less sensitive to DNA damage and oxidative stress, the efflux drug transporters MDR1 and MRP1 were not up-regulated, and doxorubicin accumulated in the cytoplasm rather than in the nucleus. Both the autophagy inhibitor chloroquine and extracellular vesicle (EV) release inhibitor GW4869 enhanced doxorubicin activity and counteracted doxorubicin resistance. In conclusion, this study identifies common modulated antigens in HRSdx cells, the associated cross-resistance patterns, and new potential therapeutic options to enhance doxorubicin activity and overcome resistance.

## 1. Introduction

In cHL rare tumor cells, the so-called Hodgkin and Reed–Sternberg (HRS) cells express high levels of CD30, CD40, IRF4, CD15, and a constitutive active nuclear factor kappa B (NF-κB) [1]. They are embedded in a rich extracellular matrix [2] and are surrounded by an immunosuppressive and protective tumor microenvironment (TME) [3], predominantly composed of inflammatory cells, including macrophages, CD4^+^ and CD8^+^ T cells, plasma cells, eosinophils, other immune cells, and stromal cells [4]. Indeed, HRS cells express and secrete immunosuppressive molecules, which can recruit and hijack normal cells to become immunosuppressive M2 tumor-associated macrophages (M2-TAMs), exhausted/anergic T-cells [4], and protumorigenic cancer-associated fibroblasts (CAFs) [5]. Thus, HRS cells can be protected by the cytotoxic effects of chemotherapy by soluble factors, extracellular vesicles, and by direct contact with the hijacked inflammatory and stromal cells of the TME [3].

Most patients with classical Hodgkin lymphoma (cHL) can be successfully treated with conventional-dose chemotherapy and radiotherapy (RT). Although chemotherapy has a high response rate, 30% of cHL patients will relapse [6] due to drug resistance, which can be either intrinsic, acquired during drug treatment [7,8,9], or promoted by the interactions with the TME [3]. The first-line therapies for cHL patients are the multidrug regimen ABVD (adriamycin, bleomycin, vinblastine, dacarbazine) or BEACOPP (bleomycin, etoposide, adriamycin, cyclophosphamide, vincristine, procarbazine, and prednisone) [6]. Both regimens include doxorubicin (adriamycin), one of the most effective anticancer agents, even though its activity can be reduced by drug resistance and its positive effects compromised by cardiotoxicity [10]. Therefore, to plan for the successful and less toxic use of doxorubicin, it would be worth investigating the characteristics of doxorubicin-resistant cells. By doing this we could find prognostic biomarkers to identify doxorubicin-resistant patients, predict therapy responses, avoid the use of ineffective chemotherapy agents, and find new/alternative drugs or drug combinations to overcome doxorubicin resistance. 

Our aim was to find common markers and mechanisms to predict and overcome doxorubicin resistance in cHL cells. To achieve our goal, we evaluated phenotypic and functional features of two cHL cell lines with acquired resistance to doxorubicin, KM-H2dx and HDLM-2dx [11], generated in our lab from KM-H2 and HDLM-2 cHL cell lines, respectively. 

We analyzed and compared, in parental cell lines (Hodgkin Reed Sternberg, HRS) and their doxorubicin-resistant counterparts (called HRSdx), survival and cancer stem cells factors; molecules involved in the cross-talk with the TME; immunosuppressive molecules; cytokines/chemokines; immunosuppressive tumor-education of monocytes; cross-resistance to anticancer drugs, including the liposomal formulation of doxorubicin caelyx and γ-radiation; the expression of drug transporters; doxorubicin intracellular distribution; DNA damage and resistance to oxidative stress; and the role of autophagy and extracellular vesicles.

## 2. Materials and Methods

### 2.1. Drugs and Inhibitors

Doxorubicin (Hikma, Pavia, Italy), PEGylated liposomal doxorubicin caelyx (Janssen Cilag, Milano, Italy), bleomycin (Sanofi, Varese, Italy), cisplatin (Accord, Monza, Italy), dacarbazine (Medac, Roma, Italy), trabectedin (PharmaMar, Milano, Italy), bendamustin (Hikma), gemcitabine (Actavis, Milano, Italy), and vinblastine (Velbe, EG Stada, Dresden, Germany) (EG) brentuximab vedotin (BV) (Adcetris, Takeda, Roma, Italy) were surplus drugs from the clinical pharmacy of CRO Aviano. Prof. Kazuo provided dehydroxy-methylepoxyquinomicin (DHMEQ) [12]. Monometil auristatine E (MMAE) and chloroquine were from Sigma-Aldrich (Milano, Italy), and GW4869 was from SelleckChem (Roma, Italy). Antibodies used in flow cytometry and Western blotting are given in Appendix A, respectively. Other reagents are described within individual protocols.

### 2.2. Cell Culture and Conditioned Media

The cHL-derived cell lines KM-H2, HDLM-2, and L-428 were from DSMZ (Braunschweig, Germany), and human monocytic THP-1 cells were from the American Type Culture Collection (ATCC, Manassas, VA, USA). KM-H2 cells (CD2/B-like) were established from the pleural effusion of a 37-year-old man with mixed cellularity progressing to lymphocyte depletion (stage IV at relapse) [13]. HDLM-2 cells (T-like), established from the pleural effusion of a 74-year-old man with nodular sclerosis, stage IV in 1982 [13], are intrinsically resistant to BV [14,15]. Here, KM-H2 and HDLM-2 cells were collectively called “HRS cells”, and doxorubicin-resistant cell lines, which we called KM-H2dx and HDLM-2dx, were collectively referred to as “HRSdx cells”. HRSdx cells were generated through the continuous exposure of HDLM-2 and KM-H2 cells to increasing concentrations of doxorubicin (10 nM to 100 nM) during sequential passages, for about 12 months. Cell lines were authenticated in our laboratory using the PowerPlex 16 HS System (Promega, Milano, Italy) and GeneMapper ID version 3.2.1 to identify DNA short tandem repeats. They were routinely tested for mycoplasm with negative results. HRS, HRSdx, and THP-1-monocytes were cultured in RPMI-1640 medium containing 10% fetal bovine serum (Gibco, Thermo Fisher Scientific, Milano, Italy) (complete medium). To maintain doxorubicin resistance, HRSdx were continuously exposed to doxorubicin: KM-H2dx cells were maintained in 20 ng/mL doxorubicin and HDLM-2dx in 30 ng/mL. HRSdx cells maintained their resistance to doxorubicin for at least two weeks in the absence of the drug. 

To obtain the conditioned medium (CM), cells were seeded at 2.0 × 10^5^ cells/mL in complete medium. After 72 h, cells were counted and CM was collected to evaluate cytokine levels and used to tumor “educate” monocytes. Cytokine release in CM was quantified using ELISA kits for CCL5, FGF-2, IL-6, IL-13, M-CSF, CCL17/TARC (all from Immunological Sciences, Rome, Italy), TGF-β1 (Invitrogen, Thermo Fisher Scientific), prostaglandin E2 (Human PGE2) (FineTest, Milano, Italy), and L-lactate (Cell Biolabs, Inc., Bergamo, Italy).

### 2.3. Cellular Assays 

To determine treatment cytotoxicity and to perform drug combination studies, tumor cells (2.0 × 10^5^ cells/mL) were seeded in 24-well plates in complete medium and treated with increasing concentrations of drugs in triplicate for 72 h. Viable cells were counted using the trypan blue dye exclusion assay. Half maximal inhibitory concentrations (IC_50_s) were calculated using CalcuSyn software, v2.1 (Biosoft, Ferguson, MO, USA) [16]. Resistance factor (RF) was calculated as the ratio of the IC_50_ of the resistant cell line (KM-H2dx and HDLM-2dx) to that of parental cell line (KM-H2 and HDLM-2). The doubling time (DT) (in days) was calculated using the following formula: = h × ln(2)/ln(c2/c1), where c is the number of cells at each time of collection and ln is a neperian logarithm (Roth V., 2006 http://www.doubling-time.com/compute.php) [17]. To test basal clonogenic growth, tumor cells were suspended at a density of 5.0 × 10^3^ cells/mL in RPMI-1640 medium, 15% FBS, and 0.8% methylcellulose and seeded in 100 µL aliquots (8 replicates) in 96-well flat-bottomed microplates. After 14 days of incubation, plates were observed under phase-contrast microscopy, and aggregates with ≥40 cells were scored as colonies [11].

To compare the effects of the conditioned medium from tumor cells on monocyte immunosuppressive differentiation, THP-1-monocytes [18] (1.0 × 10^5^ cells/mL) were cultured in complete medium (RPMI 10% FCS) supplemented with 20% CM from HRS cells and from HRSdx cells for 6 days (THP-1-monocytes become tumor-educated monocytes, E-monocytes) (CM addition every two days, 3 additions). Flow cytometry was used to assay CD206, PDL-1, and IDO expression in THP-1-monocytes.

### 2.4. Drug Combination Studies with Chloroquine and GW4869

To evaluate the effects of autophagy and extracellular vesicles (EVs) on doxorubicin activity, tumor cells were cultured with non-toxic concentrations of chloroquine (CQ) (inhibitor of autophagy) and GW4869 (inhibitor of EV release). To find the not toxic concentrations, tumor cells were cultured with increasing concentrations of CQ (0–10 µM) or GW4869 (0–10 µM) (Appendix A). Then, HRS and HRSdx cells (2.0 × 10^5^ cells/mL) were cultured with a non-toxic concentration of CQ (2.5 µM) or GW4869 (2 µM) alone, or in combination, with different concentrations of doxorubicin. After 72 h, viable cells were evaluated with a trypan blue dye exclusion assay. 

### 2.5. Doxorubicin Accumulation

Doxorubicin accumulation and distribution were evaluated via confocal microscopy and flow cytometry, owing to its red fluorescence properties. For confocal microscopy, cells (2.0 × 10^5^ cells/mL) were plated on poly-lysine 35 mm coated wells and treated with 1 µg/mL doxorubicin. Doxorubicin distribution was monitored using a confocal microscope in time-lapse xyzt acquisition mode (Leica DM IRE2) for 2 h keeping cells at 37 °C and with 5% CO_2_. For flow cytometry assays, cells (2.0 × 10^5^ cells/mL) were incubated for 2 h with different concentrations of doxorubicin (0–200 ng/mL). Then, red fluorescence intensity was evaluated on a BD FACSCanto II flow cytometer.

### 2.6. Flow Cytometry

HRS cells were stained with a panel of antibodies (Appendix A). To evaluate viability, cells were stained for 15 min with FITC Annexin-V (Thermo Fisher Scientific) and 7AAD (BD Pharmingen, Milano, Italy). For IRF4, DDR1, B-cell lymphoma/leukemia-2 (Bcl-2), and Bcl-2-associated X protein (Bax) evaluation, cells were fixed and permeabilized using the FIX & PERM Cell Fixation & Cell Permeabilization Kit (Life Technologies, Monza, Italy). To evaluate mitochondrial reactive oxygen species (mit-ROS) generation, cells were stained with 5 μM MitoSOX Red Mitochondrial Superoxide Indicator (Thermo Fisher Scientific) in working solution for 30 min at 37 °C. To detect cytoplasmic ROS (cyt-ROS), the CM-H_2_DCFDA (Thermo Fisher) dye was used. The enzymatic activity of ALDH-1 was measured with an ALDEFLUOR™ Kit for ALDH Assays (Stem Cell Technologies, Inc., Cambridge, MA, USA), as previously described [19]. Results were detected via flow cytometry on a BD FACSCanto II flow cytometer. Data were analyzed using BD FACSDiva v.8.0.1 software (BD Biosciences, Milano, Italy) unless otherwise indicated.

### 2.7. Gamma-Radiation Treatment and Cell Cycle Assay

Cells (2.0 × 10^5^ cells/mL) were seeded in 24-well plates and irradiated with photon beams of 6 MV of energy delivered by a radiotherapy linear accelerator (3–6–12 Gy). The clonogenic growth assay was performed immediately after irradiation. Cell cycle analysis was performed after 24 h from irradiation. For cell cycle analysis, cells were fixed in cold 70% ethanol for 15 min and stained with propidium iodide (PI) solution (50 µg/mL PI, 0.1% NP-40, 100 µg/mL PureLink RNase A, 0.1% sodium citrate) for 1 h. The distribution of cells in different cell cycle phases was quantified using ModFit LT 4.0 software (Verity Software House, Topsham, ME, USA). Cell viability was evaluated after 72 h of irradiation with Annexin-V/7AAD staining.

### 2.8. Real Time-PCR

Total RNA was isolated from cells using TriZol reagent (Thermo Fischer) following the manufacturer’s instructions. One microgram of total RNA was retro-transcribed using random hexamers and the AMV reverse transcriptase (Promega). One-tenth of the obtained cDNAs was amplified using primers for the following: -human MDR1 (forward 5′-TAATGCCGAACACATTGGAA-3′ and reverse 5′-TCTTCACCTCCAGGCTCAGT-3′),-human MRP1 (forward 5′-TGCAGAAGGCGGGGAGAACCTC-3′ and reverse 5′-GTCGTCCGTTTCCAGGTCCACG-3′),-human CTR-1 (forward 5′-GGGGATGAGCTATATGGACTCC-3′ and reverse 5′-TCACCAAACCGGAAAACAGTAG-3′),-human GAPDH (forward 5′-GAAGGTGAAGGTCGGAGTC-3′ and reverse 5′-GAAGATGGTGATGGGATTTC-3′).

The cRNA was then retro-transcribed using the GoScript reverse transcriptase (Promega). Quantitative real-time PCR analyses were performed using the CFX96 TM real-time PCR detection system (Bio-Rad Laboratories, Inc., Hercules, CA, USA).

### 2.9. Western Blotting

Whole-cell lysates were prepared using cold RIPA buffer (150 mM NaCl, 50 mM tris-HCl (pH 8), 0.1% SDS, 1% Igepal, and 0.5% desoxycholate sodium) containing a protease inhibitor cocktail (Roche Diagnostics S.p.a., Milan, Italy), phosphatase inhibitors, 1 mM Na3VO4, and 1 mM NaF (Sigma Aldrich, Milano, Italy). Protein concentrations were determined using the Protein Assay Dye Reagent Concentrate (Bio-Rad Laboratories, Segrate, Italy). Equal amounts of proteins were mixed with Laemmli buffer, separated using 4–20% SDS-PAGE (Criterion Precast Gel, BioRad, Milano, Italy), and blotted onto a nitrocellulose membrane (Amersham, GE Healthcare, Milano, Italy). Membrane strips were blocked with EveryBlot Blocking Buffer (BioRad), incubated at 4 °C overnight with primary antibodies (Appendix A), probed with the appropriate secondary antibodies (Appendix A), and developed using Immobilon Western Chemioluminiscent HRP Substrate. Images were acquired using a ChemiDoc XRS system (Bio-Rad).

### 2.10. Migration/Invasion Assays

These assays were performed in Boyden chambers (Sigma-Aldrich). Prior to performing migration, the lower side of the chamber was coated with either 20 μg/mL fibronectin (Sigma-Aldrich) (migration) or 50 μg/mL Growth Factor Reduced (GFR) Basement Membrane Matrix (Matrigel, Corning, Turin, Italy) (Invasion). Cells were labeled with the lipophilic CellTracker CM-DiI dye according to the vendor’s instructions (Thermo Fisher Scientific). Then cells, were seeded in 150 μL of serum-free medium in the upper sides of Boyden chambers. Each insert contained 100,000 tumor cells. The lower side contained 700 μL of complete medium. Transmigrated cells were detected using a computer-interfaced GeniusPlus microplate reader (Tecan, Milano, Italy). Migration was expressed as the percentage of migrated cells at different time points.

### 2.11. NF-kB p65 Transcription Factor Assays and TrxR

To measure the NF-kB p65 active form, tumor cells (5.0 × 10^6^ cells) were collected and nuclear protein was extracted. Briefly, cells were lysed with buffer A (10 mM HEPES (pH 7.9), 10 mM KCl, 1.5 mM MgCl_2_, 0.5 mM dithiothreitol, 0.05% NP-40, 0.5 mM PMSF, 1 mM Na_3_VO_4_, 1 mM NaF) on ice for 30 min. Samples were centrifuged at 3000 rpm for 10 min at 4 °C. Pellets (nuclear fraction) were resuspended in ice-cold extraction buffer B (5 mM HEPES (pH 7.9), 300 mM NaCl, 0.2 M EDTA, 1.5 mM MgCl_2_, 25% (vol/vol) glycerol, 0.5 mM dithiothreitol, 0.5 mM PMSF, 1 mM Na_3_VO_4_, 1 mM NaF) and incubated on ice for 30 min. Samples were centrifuged at 13,000 rpm for 20 min at 4 °C, and the supernatant was taken as the nuclear extract. The protein concentration was determined using the Bio-Rad Bradford protein assay. NF-kB DNA-binding activity was analyzed using the Transcription Factor Kit for NF-kB p65 (Thermo Fisher Scientific). Data were normalized to those in untreated HRS cells. 

To evaluate thioredoxin reductase (TrxR) activity, cells were lysed in 50 mM Tris-HCl pH 7.6, 0.1% Triton X-100, and 0.9% NaCl. TrxR (EC 1.8.1.9) was assayed using the Thioredoxin Reductase Assay Kit (Sigma-Aldrich). Enzyme activity was determined reading the absorbance at 412 nm using a spectrophotometer (Biomate 3 Thermo Spectronic, Thermo Electonic Corporation, Monza, Italy). The enzymatic activity was normalized to the protein concentration, determined using the Bio-Rad Bradford protein assay.

### 2.12. Statistical Analyses

Statistical analysis was carried out using GraphPad Prism version 6.0 software (GraphPad, La Jolla, CA, USA). Student’s *t* tests were used to compare two groups. One-way ANOVA followed by the Bonferroni correction was used for multiple comparisons. One-way ANOVA followed by Dunnett’s test was used to compare each of a number of treatments with a single control. A *p*-value < 0.05 was considered significant.

## 3. Results

### 3.1. Characterization of Doxorubicin-Resistant Cell Lines

KM-H2dx and HDLM-2dx cells had a doxorubicin IC_50_ 9- and 8-fold higher than parental cells, respectively (Table 1) [11,20]. They were bigger in size and showed an increased number of giant polynucleated cells (Figure 1A,B) with respect to parental cells. 

HRSdx cells grew more slowly than parental HRS cells, as revealed by the decreased doubling time (DT) (Figure 1C), formed fewer colonies in methylcellulose (clonogenic growth) (Figure 1D), and had a slight reduction in NF-KB activity (Figure 1E). HRSdx cells demonstrated increased migratory abilities towards complete medium (Figure 1F) and invasive potential through Matrigel-coated membranes (Figure 1G) and expressed higher levels of the motility-regulatory factors Rock and RHOA [21] (Figure 1H) and the chemokine receptor CCR5 and also CXCR4 in KM-H2dx cells (Figure 1I). These results indicate that doxorubicin-resistant cells show morphological changes, reduced proliferation, and increased migratory capabilities.

### 3.2. Survival Factors and Putative Cancer Stem Cells Markers

HRSdx cells had increased expression of survival factors, detoxifying agents, and markers identifying putative cancer stem cells (CSCs), known to exhibit drug-resistant features. HRSdx cells showed decreased CD71 and increased Bcl-2 levels with respect to their parental cell lines (Figure 2A). Bcl-xL and Jagged1 were increased only in KM-H2dx cells, while Jagged1, IRF4, and Notch-1 were slightly decreased in HDLM-2dx cells (Figure 2A). 

We also analyzed the expression of ALDH1, up-regulated in putative cHL-CSCs, cytoplasmic reactive oxygen species (cyt-ROS), down-regulated in cHL-CSCs [22,23], and mitochondrial-ROS (mit-ROS), up-regulated in CSCs of different tumor types [24]. We found that ALDH1 was expressed and up-regulated only in KM-H2dx cells (from 1% to 3% positivity) (Figure 2B), whereas cyt-ROS and mit-ROS were down- and up-regulated in both HRSdx cells, respectively (Figure 2B). In conclusion, HRSdx cells expressed putative CSCs markers and increased levels of anti-apoptotic molecules.

### 3.3. Interactions with the TME

Then, we evaluated molecules involved in the cross-talk of cHL cells with the TME [3,25]. The cytoplasmic discoidin domain receptor 1 (DDR1) and especially CD44 were down-regulated in HRSdx cells (Figure 2C). In KM-H2dx cells, both CD49d and CD29 did not change. CD49d was absent in HDLM-2 and HDLM-2dx cells, and CD29 decreased in HDLM-2dx cells (Figure 2C). In HRSdx cells, CD30, CD40, and CD86 did not significantly change, whereas CD80, CD54, and HLA-DR were down-regulated (Figure 2D). CD58, absent or expressed at very low levels, was increased in both HRSdx cells (Figure 2D). In conclusion, doxorubicin-resistant cells maintained high levels of CD40 and CD30 but had decreased expression of other molecules involved in the cross-talk with T-cells, fibroblasts, and extracellular matrix, excluding CD58 [26]. Results are summarized in the Venn diagrams (Figure 2E).

### 3.4. Immunosuppressive Molecules and Tumor Education of Monocytes

HRS cells can “educate” monocytes towards an immunosuppressive phenotype [4]. Thus, we evaluated if the acquisition of doxorubicin resistance could modify the immunosuppressive features of HRS cells. In HRSdx cells, the expression of HLA-G did not significantly change, whereas PDL-1 and CD83 were down-regulated and CD26 and especially COX2 were up-regulated (Figure 3A). The expression of CD137 increased and that of CD47 decreased only in KM-H2dx cells (Figure 3A). IDO was slightly induced only in HDLM-2dx cells. CD200 and CD206 were absent in HRS and HRSdx cells (Appendix A). An ELISA assay revealed that HRSdx cells secreted higher amounts of molecules involved in the immunosuppressive monocyte tumor education (IL-13, TGF-β, M-CSF, L-lactate, and PGE2), stromal cell proliferation (FGF), T-cell, fibroblast, and monocyte recruitment, and prognostic markers (CCL5 and TARC) [1] (Figure 3B and Appendix A). Since HRSdx cells secreted higher amounts of immunosuppressive molecules, they could be more prone to shape monocytes towards a pro-tumorigenic M2-TAM state, characterized by the up-regulation of CD206, PDL-1, and IDO. To test our hypothesis, untreated THP-1-monocytes (U-mon) were exposed to the conditioned medium (CM) derived from HRS or from HRSdx cells, generating educated (E-mon) and dxE-mon, respectively (Figure 3C). Flow cytometry analyses showed that dxE-mon had higher expression of CD206, PD-L1, and IDO than E-mon (Figure 3C). L-428-CM was used as a positive control of M2-TAM polarization (Appendix A). Figure 3D schematically summarizes common modifications found in HRSdx cells with respect to parental cells leading to immunosuppression (antigens, cytokines, tumor education of monocytes). Altogether, these results showed the increased immunosuppressive features of HRSdx cells.

### 3.5. Cross-Resistance Studies

Doxorubicin-resistant cells, selected during cancer therapy, can acquire features that can reduce (cross-resistance, CR) or increase (cross-sensitivity, CS) the efficacy of other drugs [27]. Therefore, predicting CR may avoid the use of inefficacious treatments.

We determined, in HRS and HRSdx cells, the IC_50_ values of drugs used in first-line treatment (ABVD, doxorubicin/adriamicin, bleomycin, vinblastine, and dacarbazine) and in relapsed/resistant cHL patients (bendamustin, gemcitabine, cisplatin, and BV). We also tested the liposomal formulation of doxorubicin caelyx [28] and γ-radiation [6]. The NF-kB inhibitor DHMEQ [29] and the marine drug trabectedin [11] were included in the screening. The IC_50_ and the resistance factor (RF) (ratio of drug-resistant cells IC_50_ over drug-sensitive cells IC_50_) for each drug were calculated. A value of RF ≤ 1 is considered cross-sensitivity (CS) and RF ≥ 1 indicates CR. RF values ranging from 1 to 2 indicate low CR, RF from 2 to 5 indicates moderate CR, and RF ≥ 5 indicates high CR (Table 1, Figure 4A). In Figure 4A, RF values for each drug were shown in ascending order. HRSdx cells had low/moderate cross-resistance to cisplatin, vinblastine, MMAE, BV, and bendamustin (Figure 4A). KM-H2dx and HDLM-2dx cells had high cross-resistance (RF > 5) to dacarbazine and gemcitabine, respectively (Figure 4A). HRSdx cells had no cross-resistance (CR ~ 1) to trabectedin and to the radio-mimetic bleomycin. HRSdx showed cross-sensitivity (RF < 1) to the NF-kB inhibitor DHMEQ (RF = 0.9) (Figure 4A). 

Brentuximab vedotin (BV, SGN-35; Adcetris^®^) is an anti-CD30 antibody conjugated via a protease-cleavable linker to the anti-microtubule agent monomethyl auristatin E (MMAE) [30]. It is the first approved agent for the salvage treatment of relapsed/refractory cHL after autologous stem cell transplantation [31]. Interestingly, HDLM-2 cells showed a remarkable intrinsic resistance to BV (HDLM-2 IC_50_ = 250 µM, KM-H2 IC_50_ = 10 µM) that was independent of the cytotoxic drug MMAE (HDLM-2 IC_50_ = 52.6 pg/mL, KM-H2 IC_50_ = 75 pg/mL) (Table 1). BV resistance and MMAE sensitivity were maintained in HDLM-2dx cells (Table 1 and Figure 4A). 

Then, we evaluated the possibility to overcome doxorubicin resistance using caelyx, the PEGylated liposomal formulation of doxorubicin [32]. The IC_50_ of caelyx was comparable in KM-H2 and HDLM-2 cells, but higher than that for free doxorubicin in all cell lines (Table 1) [32]. In KM-H2dx cells, caelyx partially reverted the resistance to doxorubicin (RF caelyx = 3 vs. RF doxorubicin = 9.1) (Figure 4A). Conversely, HDLM-2dx cells were extremely resistant to caelyx (caelyx RF = 21 vs. doxorubicin RF = 8) (Figure 4A). 

We also evaluated the sensitivity to γ-radiation in HRS and HRSdx cells. Gamma-radiation reduced tumor cell viability (Figure 4B) and clonogenic growth (Figure 4C). HDLM-2 cells were less sensitive to γ-radiation than KM-H2 and HRS cells and more sensitive than HRSdx (Figure 4B,C). Gamma-radiation (0–12 Gy) caused a dose-dependent increase in the G2 phase of the cell cycle in KM-H2 cells (from 6.5% to 75.2%), which was less evident in KM-H2dx cells (from 9.6 to 35.5%) (Figure 4D,E). In HDLM-2 cells, γ-radiation treatment caused a more consistent block in the G2 phase (from 21.2 to 85.7%), already evident at 3 Gy, but less marked in HDLM-2dx cells (from 22.7 to 43.7%) (Figure 4D,E). HRS cells showed a reduction in both the G1 and S phase, whereas in HRSdx cells, only G1 was reduced, confirming HRSdx cells being less sensitive than HRS cells to γ-radiation. Taken together, our results demonstrated that the acquisition of doxorubicin resistance leads to cross-resistance to several anticancer drugs and to less sensitivity to γ-radiation.

### 3.6. Mechanisms Involved in Doxorubicin Resistance

We more deeply investigated the possible mechanisms known to be involved in doxorubicin resistance [33] that include the following: increased expression of drug transporters [34]; different sub-cellular localization of doxorubicin [35]; increased response to oxidative stress [36]; decreased sensitivity to DNA damage [37]; autophagy [38]; extracellular vesicle (EV) release [39].

#### 3.6.1. Modulation of Drug Transporters 

Over-expression of the efflux drug transporters MDR1/ABCB1 and MRP1/ABCC1 can cause doxorubicin resistance by increasing drug efflux [33,34]. In KMH-2dx cells, mRNA levels of MDR1 and of MRP1 were slightly up-regulated (Figure 5A). In HDLM-2dx cells, MDR1 was absent [40,41] and MRP1 was slightly increased (Figure 5A). A Western blot assay confirmed the absence of MDR1 in HDLM-2 and HDLM-2dx cells and the absence of the significant modulation of MDR1 and MRP1 in HRSdx cells (Figure 5B). Consistently, with the decreased activity of cisplatin (Table 1), copper transporter CTR-1, involved in cisplatin influx [42], was down-regulated in HRSdx cells (Appendix A).

#### 3.6.2. Doxorubicin Uptake and Distribution 

Given that the reduced accumulation of doxorubicin in the nucleus can result in doxorubicin resistance, we determined the localization/distribution of doxorubicin in HRS and HRSdx cells. Using a flow cytometry assay, we found the reduced accumulation of doxorubicin (percentage of red fluorescent cells) in HRSdx cells with respect to HRS cells and in KM-H2 cells with respect to HDLM-2 cells (Figure 5C). Confocal time-lapse experiments (0–120 min) showed the time-dependent uptake and accumulation of doxorubicin in the cell membrane, cytoplasm, and finally in the nucleus of tumor cells (Appendix A). After 2 h, doxorubicin was localized inside the nucleus in HDLM-2 and KM-H2 cells, while in both HRSdx cells, it was mainly accumulated at the cell membrane and at the cytoplasmic level and only partially in the nucleus (Figure 5D). 

#### 3.6.3. DNA Damage 

Then, we evaluated the induction of DNA damage by doxorubicin. To induce DNA damage, HRS and HRSdx cells were treated for 24 h with doxorubicin (IC_90_ of parental HRS cells), and then, flow cytometry was used to assess the phosphorylation of histone H2AX (γ-H2AX), one of the most sensitive markers, to evaluate DNA double-strand breaks (DSBs) [43]. Treatment with this concentration of doxorubicin resulted in DNA-DSBs only in HRS cells (Figure 5E). In HRSdx cells, γ-H2AX increased only at higher concentrations of doxorubicin (HRSdx IC_90_) (Figure 5F). In conclusion HRSdx cells were less sensitive to DNA damage mediated by doxorubicin, likely owing to the reduced accumulation of doxorubicin in the nucleus. 

#### 3.6.4. Resistance to Oxidative Stress 

Doxorubicin resistance in HRSdx cells could be attributed to the reduced sensitivity to oxidative stress [36]. To investigate the ability of tumor cells to counteract oxidative stress, both short-term and long-term incubations with H_2_O_2_ have been exploited [44]. A short incubation (1 h) with H_2_O_2_ decreased, in a dose-dependent manner, the viability of HRS cells but only slightly affected HRSdx cells (Figure 6A,B). Also, after a 24 h treatment with H_2_O_2_, HRS cells were more sensitive than HRSdx cells, and KM-H2 cells were more sensitive than HDLM-2 cells (Figure 6C). Consistently, HRSdx cells had higher levels of the detoxifying enzyme thioredoxin Reductase (TrxR) (Figure 6D). 

#### 3.6.5. Inhibition of Autophagy and EV Release

Autophagy and EVs can modify doxorubicin activity [38,39]. Autophagy, involved in HRS cell survival and growth [45], is a self-degradative process that removes unnecessary or dysfunctional components through a lysosome-dependent mechanism [38]. Its inhibition by chloroquine (CQ), which blocks the fusion of autophagosomes with lysosomes and promotes the accumulation of degraded/misfolded proteins, could enhance doxorubicin cytotoxicity [38]. We found that CQ, used at non-toxic concentrations (Appendix A), enhanced doxorubicin activity in HRS cells and overcame doxorubicin resistance in HRSdx cells. CQ strongly reduced the HRSdx doxorubicin IC_50_ getting closer to that of parental cells (Figure 7A). 

Recent research suggests a key role in drug resistance for extracellular vesicles (EVs), small membrane-bound vesicles that transfer cargo molecules, including drugs, in the extracellular space, thus reducing drug intracellular accumulation [39]. To investigate a possible role of EVs in the acquired doxorubicin resistance, we cultured tumor cells with GW4869, an inhibitor of EV release [46]. We found that a non-toxic concentration of GW4869 (Appendix A) enhanced doxorubicin cytotoxicity in HRS cells and overcame doxorubicin resistance in HRSdx cells (Figure 7B). Taken together, our results suggest that both autophagy and EVs protect HRS cells from doxorubicin toxicity. 

## 4. Discussion

For cHL therapy, the main issue is to select patients that could benefit from a standard frontline intensive chemotherapy from those that are poor responders and would take advantage of new alternative therapies. 

Drug resistant cHL-derived cell lines could help to find new predictive factors for prognosis and response to drug treatment and more novel active and less toxic therapeutic strategies [47].

Here, we described the functional and phenotypic characteristics of KM-H2dx and HDLM-2dx cell lines with acquired resistance to doxorubicin. They were generated in our lab via continuous exposure to doxorubicin in the cHL cell lines KM-H2 and HDLM-2, respectively [13]. To our knowledge, this is the first report analyzing the characteristics of HRS cells with acquired resistance to doxorubicin.

HRSdx cells, compared with parental HRS cells, showed a higher number of giant mostly multinucleated cells; reduced doubling time and clonogenic growth capacity; increased Bcl-2 and CSCs markers; increased migration and invasive capabilities; increased CCR5 expression; decreased levels of molecules involved in the cross-talk with the TME (CD44, DDR1, CD54, CD80, HLA-DR), but increased CD58; increased secretion and expression of molecules involved in TME formation (CCL5 and TARC) and immunosuppression (PGE, TGF-β, CD26, COX-2); increased capability to induce the M2-TAM phenotype in monocytes; low or moderate cross-resistance to vinblastine, bendamustin, cisplatin, dacarbazine, gemcitabine, BV, MMAE, and γ-radiation; no cross-resistance to bleomycin, trabectedin, and DHMEQ; CTR-1 down-regulation; decreased accumulation of doxorubicin in the nucleus and consequent decreased DNA damage; decreased sensitivity to oxidative stress and increased TrxR enzymatic activity; and enhanced sensitivity to doxorubicin used in combination with inhibitors of autophagy and EV release.

Moreover, only KM-H2dx cells had increased expression of ALDH1, Bcl-xL, CXCR4, Jagged1, and CD137; only HDLM-2dx cells acquired remarkable collateral resistance to caelyx.

HRSdx cells expressed markers of the putative CSCs: low levels of intracellular ROS, increased ALDH1, a decreased doubling time [22,48] and cyt-ROS decrease, and higher expression of mit-ROS [24]. We could speculate that doxorubicin treatment selects for and expands HRS cells more prone to survive and to be less sensitive to anticancer drugs. 

Consistently with the increased migration and invasion capability, HRSdx cells expressed higher levels of RhoA and ROCK [21,49] and the chemokine receptors CCR5 and CXCR4 [50] and lower levels of several molecules involved in the interactions with the TME. Genetic alterations could be the reason of the absence of CD49d in HDLM-2 and HDLM-2dx cells [1,51]. 

In HRSdx cells, the decrease in HLA-DR expression could favor the escape from immune system recognition and the increase in CD58, by improving the interactions with CD40L+CD2+ rosetting T cells, and could support tumor cell survival [26]. Thus, doxorubicin-adapted HRS cells, with increased invasive abilities and the reduced expression of molecules involved in the cross-talk with the TME could be more prone to evade from immune system recognition, escape from the lymph node, and metastasize.COX-2 over-expression by tumor cells is an adverse independent prognostic factor in cHL patients treated with ABVD [52]. Indeed, chemotherapy was found to induce COX-2/PGE2 up-regulation in different cancer models and to decrease the effects of the combination of chemotherapy with immunotherapy [4]. COX-2 was up-regulated in both HRSdx cells. Thus, we can speculate that its association with poor prognosis could be related not only to its immunosuppressive activity but also to the development of drug resistance, indicating COX-2 as a potential marker of HRS acquired doxorubicin resistance.

The doxorubicin-adapted HRS cells secreted increased levels of cytokines/chemokines involved in TME formation (CCL5, TARC, M-CSF, and FGF) and immune-suppression (PGE2, IL-13, TGF-β, and L-lactate) [4]. Consistently, HRSdx-CM was more effective than HRS-CM in inducing an immunosuppressive M2-like TAM phenotype in monocytes. They up-regulated PDL-1, IDO, and especially CD206, known to promote matrix-remodeling and lymphoma dissemination [2]. Thus, HRS with acquired resistance to doxorubicin cells could be more efficient in building an immunosuppressive TME and hijacking monocytes [4]. 

Cross-resistance studies showed that the acquisition of doxorubicin resistance by HRS cells was associated with modest/intermediate collateral resistance to several other chemotherapy agents. However, we found that bleomycin, which is part of ABVD treatment, the NF-kB inhibitor DHMEQ [29], and trabectedin had comparable activity in HRS and HRSdx cells [11]. Trabectedin, able to counteract HRS/TME interactions [53] and to enhance the anticancer activity of the inhibitor of telomerase activation BIBR153 in HRS cells [54], could be a promising option for patients with refractory/resistant cHL.

To increase doxorubicin accumulation in tumor cells and especially to overcome/reduce its cardiac toxicity, the PEGylated liposomal formulation of doxorubicin (caelyx) has been proposed in relapsed or refractory cHL [55]. While in KM-H2dx, doxorubicin resistance was partially reverted by caelyx, HDLM-2dx developed a strong resistance to caelyx, which could be attributed to the defective uptake and internalization of liposomes or to the exosome-mediated expelling of the drug, as demonstrated in U937 cells [56]. Chen R and colleagues [8] demonstrated that acquired BV resistance was associated with the over-expression of the MDR1 drug efflux transporter. However, we found that HDLM-2 and HDLM-2dx cells, intrinsically resistant to BV but not to MMAE, did not express MDR1, indicating different mechanisms involved in intrinsic or acquired BV resistance and suggesting different approaches to counteract them. Further studies are needed to discover the molecular mechanisms of the cross-resistance to caelyx observed in HDLM-2dx cells and to BV in both HDLM-2 and HDLM-2dx cells. 

HRSdx cells are less sensitive than parental cells to the cytotoxic effects of γ-radiation suggesting that the adaptation of HRS cells to doxorubicin could also decrease the efficacy of radiation therapy in cHL patients [57]. 

Several mechanisms are involved in the acquired resistance to doxorubicin [33]. One such mechanism is usually associated with the increased expression of the drug efflux transporters [34]. We found only a slightly increase in the mRNA levels of MRP1 in HRSdx cells and of MDR1 in KM-H2dx cells [9], outlining the poor role of these drug transporters in the acquired doxorubicin resistance in cHL tumor cells. However, consistently with the cross-resistance to cisplatin, the copper transporter CTR-1, involved in cisplatin influx [42], was down-regulated in HRSdx cells. 

The decreased localization of doxorubicin in the nucleus [35], together with the increased expression of anti-apoptotic molecules Bcl-2 and Bcl-xL, putative CSCs markers, and detoxifying enzyme TrxR, could explain the reduced sensitivity of HRSdx cells to DNA damage and oxidative stress. 

Finally, we found a new role of autophagy and EVs in cHL. Autophagy, required for HRS cell survival [45], could regulate doxorubicin activity by decreasing the accumulation of degraded proteins or by promoting drug sequestration and degradation into lysosomes [58]. Consistently, we found that the autophagy inhibitor CQ enhanced doxorubicin activity and overcame doxorubicin resistance in HRSdx cells, likely owing to the increased autophagic activity of HRSdx cells.

EVs can reduce drug activity in different cancer models and are used as biomarkers of resistance to therapy [39]. Indeed, we found that the EV inhibitor GW4869 enhanced doxorubicin cytotoxicity in HRS cells and counteracted doxorubicin resistance in HRSdx cells, suggesting that drug resistance could be mediated, at least in part, by the EV-mediated expelling of doxorubicin leading to its decreased intracellular accumulation. Given that GW4869 was found to increase the activity of PEGylated liposomal doxorubicin in U937 cells [56], we can speculate that an increase in the EV-mediated expelling of caelyx could be the reason for the high cross-resistance to caelyx observed in HDLM-2dx cells. Indeed, cHL-derived EVs could be involved not only in the modification of the secretome of fibroblasts toward a CAF phenotype [5], but also in the resistance to chemotherapy. Moreover, EVs that were detected in circulating plasma from pediatric cHL [59] could be a suitable source of new biomarkers of drug resistance [60].

## 5. Conclusions

Therapeutic approaches to overcome drug resistance and the discovery of new prognostic and drug resistance markers are central goals in cHL relapsed patients. 

Here, we demonstrated that the acquisition of doxorubicin resistance decreased the sensitivity to several anticancer agents and enhanced the immunosuppressive abilities of HRS cells. We found several molecules with modified expression and secretion in HRSdx cell lines, like immunosuppressive cytokines and COX2, which could predict the acquisition of doxorubicin resistance and the consequent cross-resistance to other anticancer drugs and radiation therapy.

Moreover, HDLM-2 cells could be a useful tool to study the intrinsic resistance to BV and HDLM-2dx to study the resistance to caelyx, the liposomal formulation of doxorubicin. 

Given that in HRS and HRSdx cells, doxorubicin activity was enhanced by inhibitors of autophagy and EV release, their clinical use could be repurposed to overcome doxorubicin resistance in relapsed cHL patients. Also trabectedin, with no cross-resistance to doxorubicin, could be a treatment option for relapsed cHL patients. 

A deeper investigation of EVs secreted by HRS and HRSdx cells or detected in the plasma of cHL patients could increase our knowledge about TME interactions, clarify mechanisms promoting drug resistance, or be used to monitor the status of cHL patients.

In conclusion, some molecules detected in HRS cells or in plasma after doxorubicin treatment may represent markers of drug-resistance acquisition, predict poor prognosis, and help to find new therapeutic options for relapsed/resistant cHL patients. 

## Figures and Tables

**Figure 1 cells-12-02732-f001:**
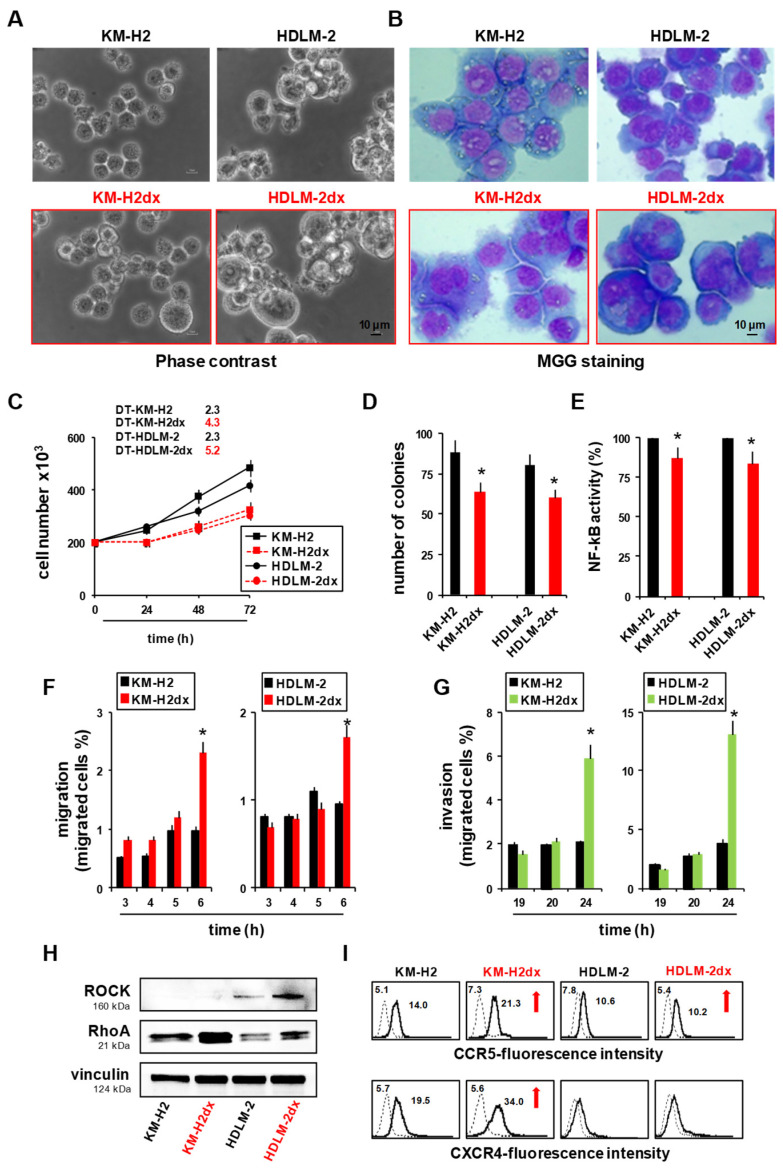
Characteristics of HRS and HRSdx cells. (**A**) Phase-contrast photo-micrographs of HRS and HRSdx cells. (**B**) Morphological images of HRS (upper panels) and HRSdx cells (red, lower panels) obtained after May–Grünwald–Giemsa staining (MGG). (Magnification, ×20; scale bar, 10 µm.) (**C**) Growth curves of HRS and HRSdx cells. The number of viable cells was evaluated via trypan blue dye exclusion assays. The calculated doubling times (DTs, in days) for each cell line were reported in the figure. (**D**) Clonogenic growth. Cells were seeded in medium containing 0.8% methylcellulose. After 14 days, aggregates with ≥40 cells were scored as colonies. Values (total number of colonies) are the mean ± SD of eight replicates of three independent experiments. (**E**) HRS and HRSdx cells were lysed, and NF-kB p65 transcription factor activity was analyzed in nuclear extracts using the Transcription Factor Kit (p65). Results are represented as the percent of control (activity HRSdx respect to HRS parental cells) and are the mean ± SD of three independent experiments. Chemotaxis assays in Boyden chambers. (**F**) Migration of HRS and HRSdx cells through fibronectin-coated (20 µg/mL) chambers towards 20% complete medium. Data are the percentages of cells that migrated from the serum-free upper chamber to the lower complete medium chamber. (**G**) Invasion. Migration of HRS and HRSdx cells through Matrigel-coated (50 ug/mL) chambers towards 20% complete medium. Data are the percentages of cells that migrated from the Matrigel-coated upper chamber to the lower complete medium chamber. (**H**) Western blot analysis for ROCK, RhoA, and vinculin in HRS and HRSdx cells. Images were acquired using a ChemiDoc XRS system (Bio-Rad). (**I**) Flow cytometry expression of CCR5 and CXCR4 in HRS and HRSdx cells. Red arrows indicate up-regulated antigens. Mean fluorescence intensities are reported in the boxes. * *p* < 0.05 HRSdx vs. HRS cells.

**Figure 2 cells-12-02732-f002:**
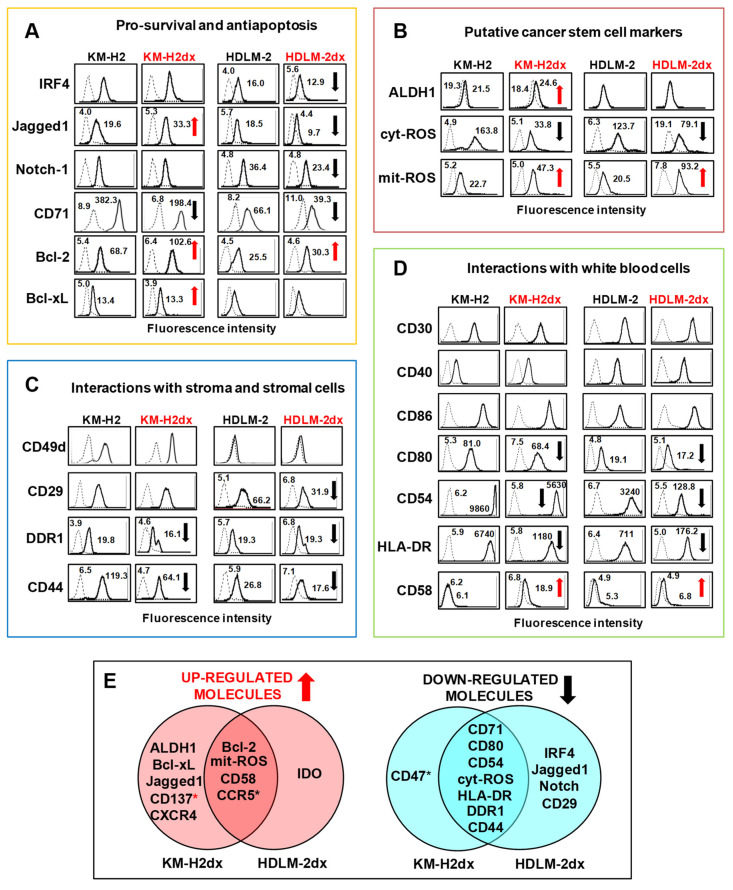
Phenotypes of HRS and HRSdx cells. Flow cytometry assay of molecules expressed by HRS and HRSdx cells. Representative flow cytometry histograms showing the expression of (**A**) survival factors and antiapoptotic molecules, (**B**) markers of the putative cancer stem cells, and (**C**) molecules involved in the interactions with collagen and stromal cells (**D**) or with white blood cells (lymphocytes, monocytes, eosinophils, and mast cells). Mean fluorescence intensities are reported in the boxes. Red arrows indicate up-regulated antigens and black arrows down-modulated antigens in doxorubicin resistant HRSdx cells with respect to parental HRS cells. (**E**) Venn diagrams showing the molecules modulated in both HRSdx cells and those specifically * up-regulated (red) or * down-regulated (blue) in KM-H2dx and HDLM-2dx.

**Figure 3 cells-12-02732-f003:**
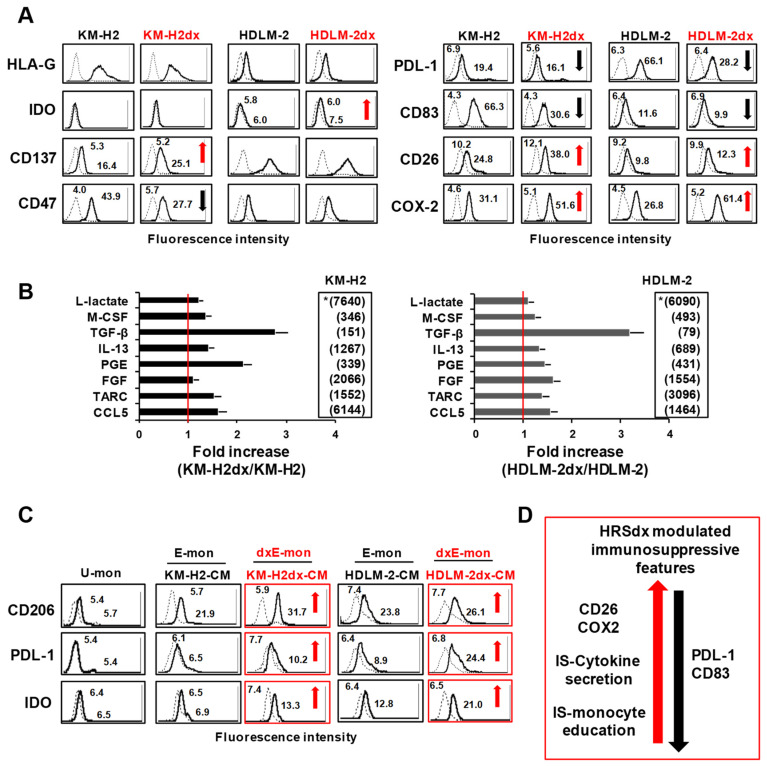
Tumor cell expression and secretion of immunosuppressive molecules and monocyte immunosuppressive education by tumor cell conditioned medium (CM). (**A**) Flow cytometry assay of immunosuppressive molecules expressed by HRS and HRSdx cells. Red arrows indicate up-regulated antigens and black arrows down-modulated antigens in HRSdx cells with respect to HRS cells. (**B**) Cytokines secreted by HRS and HRSdx cells cultured for 72 h in complete medium. Their concentrations were evaluated by an ELISA assay and reported as pg × 10^6^ cells, excluding L-lactate (*, ng/10^6^ cells). Values for KM-H2 and HDLM-2 are shown in the respective insert. Bar charts report the fold-increase in the concentration of each chemokine secreted by HRSdx cells with respect to HRS cells. (**C**) Monocytic THP-1 cells were cultured with HRS-CM and HRSdx-CM, and then, CD206, PDL-1, and IDO expression was evaluated via flow cytometry. Mean fluorescence intensities are reported in the boxes. Red arrows indicate antigens up-regulated by HRSdx-CM with respect to HRS-CM. (**D**) Immunosuppression (IS). Schematic representation of common HRSdx modifications in cells leading to immunosuppression (antigens, cytokines, monocytes, tumor education). The red arrow indicates up-regulated antigens and the black arrow down-modulated antigens (HRSdx respect to HRS cells).

**Figure 4 cells-12-02732-f004:**
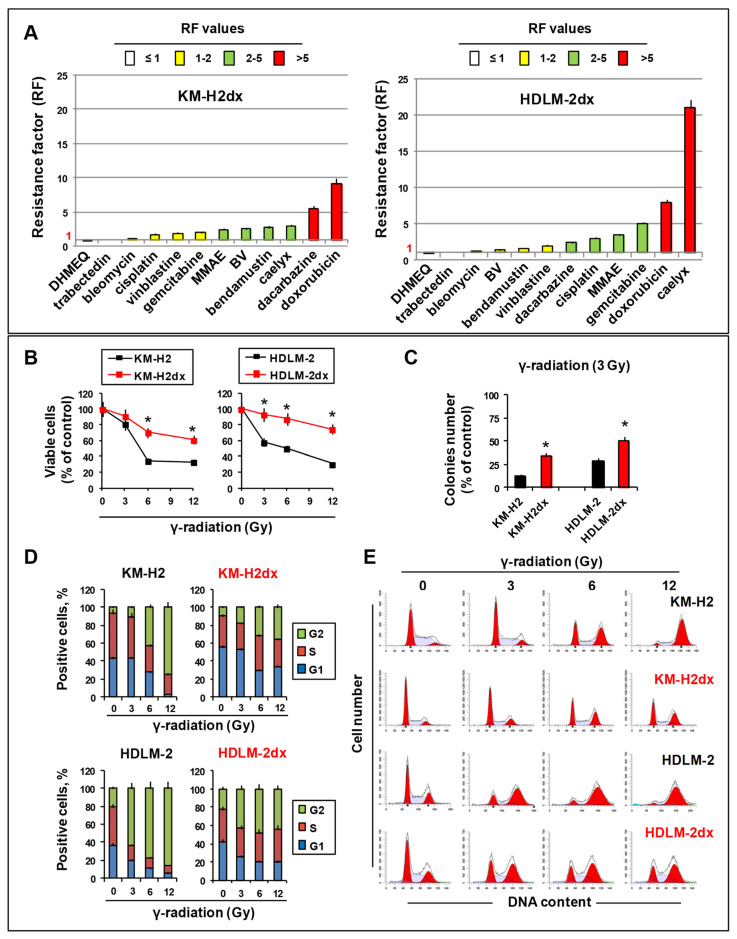
Cross-resistance pattern and γ-radiation activity in HRS and HRSdx cells. (**A**) The resistance factor (RF) value is the ratio of the HRSdx IC_50_ (KM-H2dx and HDLM-2dx) over the HRS IC_50_ (KM-H2 and HDLM-2). RFs are reported in ascending order. RF < 1 indicates cross-sensitivity (CS), and RF ≥1 indicates cross-resistance (CR). An RF ranging from 1 to 2 indicates low CR, an RF from 2 to 5 indicates moderate CR, and an RF ≥ 5 indicates high CR. (**B**–**D**) Tumor cells were treated with γ-radiation (0–12 Gy). Then, cell viability, clonogenic growth, and cell cycle distribution were evaluated. (**B**) Cells were double-stained with Annexin-V-FITC and 7AAD and analyzed via flow cytometry. Bar charts show the percentage of viable cells (Annexin-V and 7AAD negative cells). (**C**) Clonogenic growth assay. Untreated and γ-radiation-treated cells were seeded in medium containing 0.8% methylcellulose and cultured for 14 days; aggregates with ≥40 cells were scored as colonies. Values (total number of colonies) are the mean ± SD of eight replicates. (**D**) Bar charts show the percentage of cells in each cell cycle phase, evaluated after propidium iodide staining and flow cytometry analysis. (**E**) Representative cytofluorimetric histograms of the cell cycle progression after γ-radiation treatment. Results are the mean ± SD of three independent experiments. * *p* < 0.05 HRSdx vs. parental HRS cells.

**Figure 5 cells-12-02732-f005:**
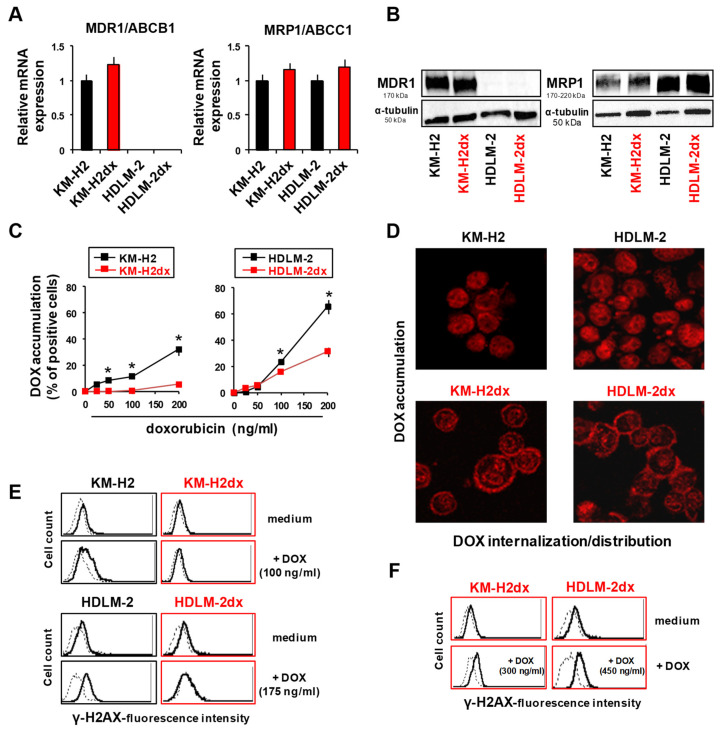
Expression of drug transporters, uptake, distribution, and DNA damage via doxorubicin in HRS and HRSdx cells. (**A**) Relative mRNA expression of MDR1/ABCB1 and MRP1/ABCC1 in HRS and HRSdx cells using GAPDH gene expression as internal control. (**B**) Western blot for MDR1, MRP1, and α-tubulin expression. (**C**) Flow cytometry-based doxorubicin accumulation assay. HRS and HRSdx cells were incubated with doxorubicin (0–200 ng/mL) for 2 h. Then, the percentage of red fluorescence-positive cells was evaluated via flow cytometry. (**D**) Cells were incubated with doxorubicin (DOX, 1 µg/mL). After 2 h, doxorubicin internalization and distribution were evaluated via confocal microscopy. (**E**) HRS and HRSdx cells were incubated for 24 h with doxorubicin (KM-H2 IC_90_ = 100 ng/mL and HDLM-2 IC_90_ = 175 ng/mL). Then, γ-H2AX expression was evaluated via flow cytometry. (**F**) KM-H2dx and HDLM-2dx cells were incubated for 24 h with doxorubicin (KM-H2dx IC_90_ = 300 ng/mL and HDLM-2dx IC_90_ = 450 ng/mL). γ-H2AX expression was evaluated via flow cytometry. Results are the mean ± SD of three independent experiments. * *p* < 0.05 HRSdx vs. parental HRS cells.

**Figure 6 cells-12-02732-f006:**
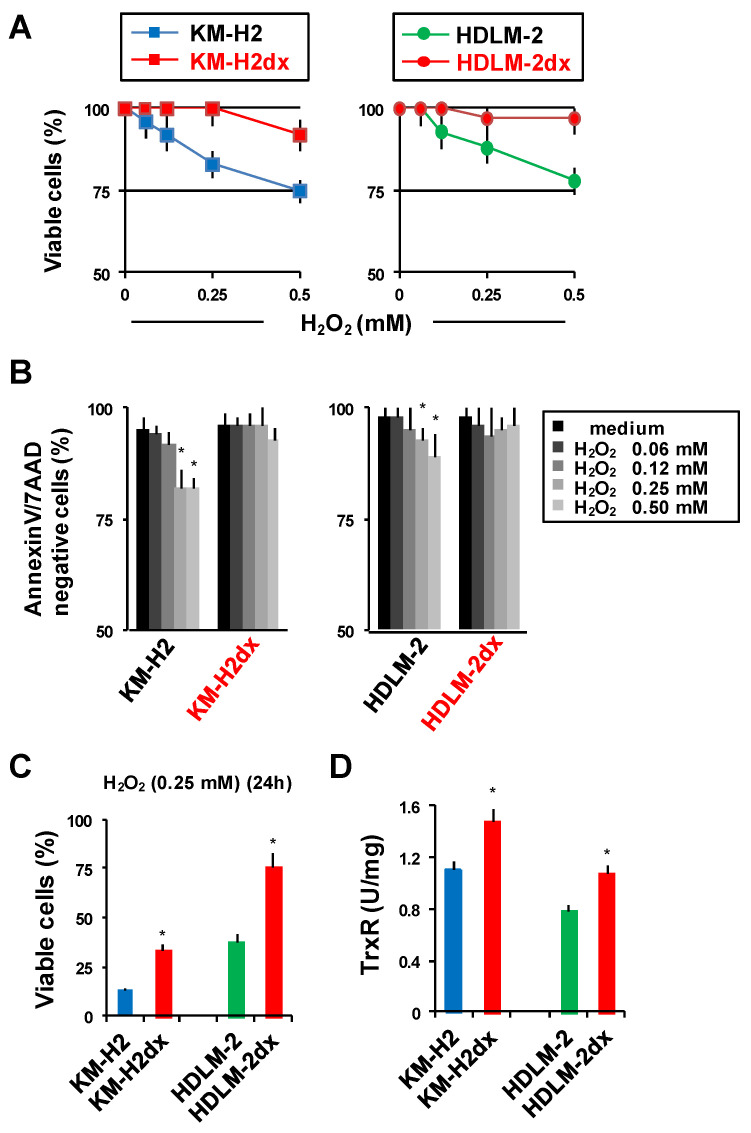
Sensitivity of HRS and HRSdx cells to oxidative stress. HRS and HRSdx cells were treated with H_2_O_2_ (0–0.5 mM). (**A**) After 1 h, cell viability was evaluated with a trypan blue dye exclusion assay. (**B**) Alternatively, cells were double-stained with Annexin-V-FITC and 7AAD and analyzed via flow cytometry. Bar charts show the percentage of viable cells (Annexin-V- and 7AAD-negative cells). (**C**) HRS and HRSdx cells were treated for 24 h with H_2_O_2_ (0.25 mM). After 24 h, cell viability was evaluated with a trypan blue dye exclusion assay. (**D**) TrxR enzymatic activity was evaluated using a TrxR assay kit and expressed as U/mg of protein. Results are the mean ± SD of three independent experiments. * *p* < 0.05 HRSdx vs. parental HRS cells.

**Figure 7 cells-12-02732-f007:**
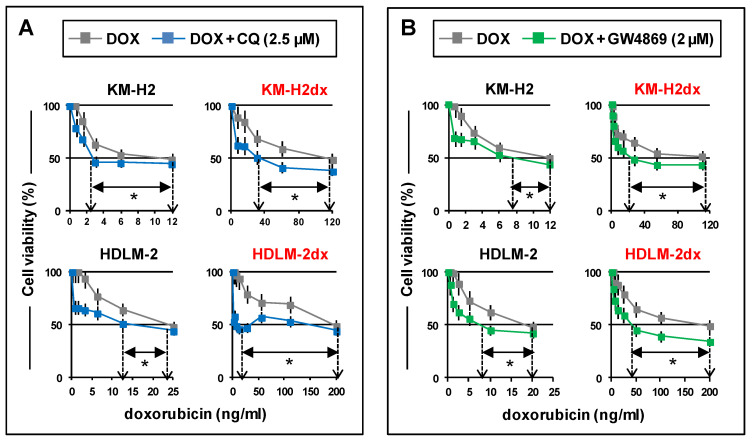
Chloroquine and GW4869 enhance cell death induced by doxorubicin. HRS and HRSdx cells were cultured with doxorubicin (DOX) in the presence or not in the presence of (**A**) a non-toxic concentration of chloroquine (CQ) (2.5 µM) or (**B**) GW4869 (2 µM). After 72 h, cell viability was evaluated via a trypan blue dye exclusion assay. Results are the mean ± SD of three independent experiments. * *p* < 0.05 DOX vs. DOX plus CQ or DOX plus GW4869. Arrows indicate the IC_50_ of doxorubicin, in the presence or not in the presence of CQ or GW4869. The difference in the IC_50_ is shown by the horizontal black double-headed arrow.

**Table 1 cells-12-02732-t001:** Cross-resistance pattern of doxorubicin-resistant HRS cells. Half-maximal inhibitory concentrations (IC_50_s) of chemotherapy agents in KM-H2, KM-H2dx, HDLM-2, and HDLM-2dx. Mean ± standard deviation (SD).

Chemotherapy Agent (IC_50_)	Cell Line			
	KM-H2	KM-H2dx	HDLM-2	HDLM-2dx
doxorubicin (ng/mL)	12 ± 0.9	110 ± 10.9	25 ± 1.9	200 ± 19
caelyx (ng/mL)	408 ± 38	1280 ± 115	480 ± 42	10,333 ± 1091
bleomycin (µg/mL)	2.5 ± 0.4	2.6 ± 0.3	10 ± 0.09	12 ± 0.15
binblastine (nM)	0.25 ± 0.03	0.49 ± 0.03	0.31 ± 0.02	0.67 ± 0.07
dacarbazine (µg/mL)	47.5 ± 5.1	265 ± 24	115 ± 10	280 ± 25
bendamustin (µM)	7.25 ± 0.06	20.2 ± 1.8	10.8 ± 1.7	17.4 ± 1.5
gemcitabine (nM)	0.53 ± 0.04	1.1 ± 0.15	3.1 ± 0.4	18 ± 1.6
cisplatin (µM)	0.48 ± 0.005	0.83 ± 0.07	1.3 ± 0.04	4 ± 0.38
brentuximabv. (µg/mL)	10 ± 1.1	25 ± 2.3	250 ± 28	350 ± 33
MMAE (pg/mL)	75 ± 7.9	182 ± 16	53 ± 5.03	183 ± 17
trabectedin (pM)	140 ± 13.5	150 ± 14	185 ± 16	187 ± 16
DHMEQ (µM)	5.1 ± 0.6	4.7 ± 0.05	4.8 ± 0.5	4.1 ± 0.35

## Data Availability

Data will be available on request.

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
