# Peer review of "In Doxorubicin-Adapted Hodgkin Lymphoma Cells, Acquiring Multidrug Resistance and Improved Immunosuppressive Abilities, Doxorubicin Activity Was Enhanced by Chloroquine and GW4869"

_cells, 2023, doi:10.3390/cells12232732_

Round 1

Reviewer 1 Report

Comments and Suggestions for Authors

The manuscript authored by Naike Casagrande et al presents a systematic investigation into the molecular patterns and cross-resistance phenomena observed in Doxorubicin-adapted Hodgkin lymphoma cells, offering valuable insights into potential therapeutic avenues to address Doxorubicin resistance.

Overall, the study is commendably designed, offering a comprehensive exploration of the characteristics of two Doxorubicin-adapted Hodgkin lymphoma cell lines. The conclusions drawn are well-founded on the current data, and the article effectively communicates the study design and scientific findings. The background information appropriately underscores the significance of the research. However, to align with the publishing standards of the journal, a few enhancements are recommended.

1.     Title: The current title may lead readers to infer that the authors successfully overcame the resistant nature of Doxorubicin-adapted Hodgkin lymphoma cells. While the manuscript thoroughly investigates the patterns in these cells and proposes potential targets to reverse Doxorubicin resistance, it lacks evidence demonstrating the reversal of the resistant feature. Consequently, the title should be revised to accurately reflect the core results.

2.     Figure 1B: Kindly incorporate a scale for clarity.

3.     Figure 1E: Include the unit for better interpretation.

4.     Figure 1H: Noteworthy is the presence of two bands for protein ROHA in the western blot. An explanation for this observation would enhance the reader's understanding.

5.     General: The fluorescence intensity figures in the manuscript could benefit from improved clarity. The current presentation lacks efficiency in conveying key information, with the decrease and increase pattern marked separately by arrows without clear statistical information. I suggest changing these figures to bar or box plots and including the raw output as supplementary information for a more straightforward interpretation.

Reviewer 2 Report

Comments and Suggestions for Authors

This manuscript studies the drug-resistance of Hodgkin lymphoma towards DOX and the mechanism of drug resistance as well as the combination therapy strategy. The whole work is well designed and organized, and the data is adequate to support the conclusion. I recommend minor revision.

1. The Introduction part should be enriched. More discussion for Hodgkin lymphoma is suggested to be given.

2. The Conlusion part is too simple. More detailed conclusion about therapeutic efficacy, mechanism is suggested to be added.
